# Changes in Psychological Challenges, Positive Experiences, and Coping Strategies among International Students in the United States before and during the COVID-19 Pandemic: A Qualitative Study

**DOI:** 10.3390/ijerph21091232

**Published:** 2024-09-18

**Authors:** Chulwoo Park, Shannon Shimada, Airi Irene Trisnadi, Patricia Angelica

**Affiliations:** 1Department of Public Health and Recreation, San José State University, San Jose, CA 95192, USA; shannon.shimada@sjsu.edu (S.S.); patricia.angelica@sjsu.edu (P.A.); 2Department of Psychology, San José State University, San Jose, CA 95192, USA; airi.trisnadi@sjsu.edu

**Keywords:** COVID-19, international student, coping, mental health, in-depth interview, qualitative study

## Abstract

The purpose of this study was to understand how the challenges and coping strategies among international students in the United States changed before and during the COVID-19 pandemic. We conducted a qualitative study through one-on-one in-depth interviews during April–June 2021. A total of 34 international students in the San Francisco Bay Area participated in this study. NVivo was used to analyze the qualitative data. Before COVID-19, the challenges were primarily related to a sense of belonging, such as culture shock, language barriers, and loss of identity. However, some positive experiences from school life and new culture were also found before COVID-19. During the pandemic, students faced difficulties with online learning and lockdown protocols that they had never experienced before. The ongoing challenges before and during COVID-19 were identity crisis, racism and discrimination, and financial struggles. Common coping strategies for overcoming these difficulties included engaging in physical activity, social exposure, self-improvement, and mental self-care. The frequency of online engagement and motivation for self-improvement were particularly highlighted during COVID-19. However, international students expressed a preference for improved support systems, including financial aid, paid research opportunities, and more accessible campus resources. Future research could focus on the strategies to support the psychological well-being of international students in coping with unexpected situations, such as disease outbreaks.

## 1. Introduction

According to the International Institute of Educational Exchange, more than 900,000 international students studied in the U.S. in 2021, with the state of California hosting the highest with over 130,000 students [1]. There had been a steady increase of international students in the U.S. for decades [2]. However, in the 2019–2020 academic year, there was a decline in enrollment of international students [2]. With the onset of the COVID-19 pandemic in late 2019, lockdown restrictions were implemented in March 2020, suspending the entry of immigrants and non-immigrants to the U.S. [3,4]. Travel restrictions prohibited foreign students from entering the U.S. and created additional complications for international students already studying in the U.S. [2]. As pandemic restrictions continued, the decline in international student enrollment persisted during the 2020–2021 academic year, with a 34.1% decrease in undergraduate enrollment and a 45.1% decrease in graduate student enrollment [5].

Before the COVID-19 pandemic, international students brought diversity to higher education but faced many challenges that domestic students did not. International students encountered difficulties when studying abroad in the U.S. due to limited resources, English proficiency, social integration, and differences in culture and academic systems [6,7,8]. For students transitioning to college, adjustments are typically made to accommodate the differences in higher education; however, when these adjustments are combined with adapting to a new country with a different language and culture, the transition becomes more difficult, leading to increased stress that affects their overall mental well-being [7,9]. Because of the transition to the U.S. education system and differences in academic expectations, international students often struggle with socialization, which impacts their psychological well-being [10]. Some students who were academically successful in their home country encounter academic adversity for the first time due to differences in the U.S. system, resulting in reduced confidence due to perfection stress [11,12,13]. Academic writing and proficiency in English pose challenges for international students, creating additional barriers to academic success and increasing their stress levels [12,14]. Many international students felt that the support from their universities was inadequate. Although most institutions had websites in place for international students, few provided direct contact information for personnel to assist them [15,16]. Moreover, student support offered by the universities was primarily limited to the initial arrival period [15]. Little support was provided to help international students transitions to the U.S. education system in areas such as classroom etiquette, student safety, linguistic support, and cultural support [13,15].

In early 2020, the COVID-19 pandemic hit, affecting college students, especially international students studying in the U.S. In response to the pandemic restrictions, international students faced an increase in both physiological and psychological risks. COVID-19 presented severe risks not only to elderly patients but also to healthy young individuals, including students who were physically active [17]. Symptoms such as fatigue, muscle weakness, and reduced respiratory function associated with COVID-19 infection were linked to an altered psychological state among college students, including post-traumatic stress disorder (PTSD), anxiety, and depression [17,18]. In addition to psychological challenges, international students were more likely to experience mental health issues related to stress, homesickness, loneliness, and fear [7,11,19]. Depression was the most common symptom reported by international students seeking help from counselors, often linked to acculturalization stress [12]. In 2020, the prevalence of major depressive disorder doubled, and generalized anxiety disorder increased by 1.5 times among international students compared to the previous year [20]. Many international students also lacked a sense of belonging, and social support is a known factor in reducing anxiety among this population [12,16]. In addition to the struggles of adapting to the U.S. higher education system, the COVID-19 pandemic introduced new challenges that further impacted their mental health [20,21]. International students from non-English-speaking countries also faced discrimination, hostility, and harassment [11,22]. To study in the U.S., international students must navigate numerous challenges, including securing an F-1 visa, attending a visa interview, and demonstrating financial stability [23,24]. Many international students continue to worry about maintaining their visas once they arrive in the US [25]. Concerns over immigration status intensified as new restrictions were introduced, increasing stress for international students who feared deportation [26,27,28].

Although nearly one million international students study in the U.S., little attention is given to their needs and psychological well-being [12,22]. Notably, few studies have explored mental health challenges both before and during the COVID-19 pandemic. This study aimed to identify the psychological challenges and positive experiences that international students encountered while studying at a public university in the San Francisco Bay Area before and during the COVID-19 pandemic. We also analyzed the changes in coping strategies due to the pandemic, the ongoing support systems in place regardless of COVID-19, and the desired support systems needed by international students to manage their mental well-being.

## 2. Materials and Methods

### 2.1. Study Design

One-on-one in-depth interviews were conducted with a total of 34 participants. The inclusion criteria were as follows: (1) Participants must have been enrolled at a university in the San Francisco Bay Area for at least one semester from Spring 2020 to Spring 2021, (2) must be 18 years old or older, (3) must be studying at the university on an F-1 visa, (4) must be able to read, write, and speak in English, and (5) must have full-time student status. We used an online survey administered through Qualtrics^XM^ (Qualtrics International Inc., Provo, UT, USA) to collect their demographic information. The in-depth interviews were structured around three main objectives. First, we aimed to understand the pre-pandemic experiences of international students, including their overall experience at their institution and the specific challenges they faced. This helped establish a baseline for comparison against experiences during the pandemic. Second, we examined the impact of COVID-19 on students’ living situations, academic performance, and additional challenges they encountered. This aimed to highlight how the pandemic altered their experiences. Third, we identified the coping strategies and support systems that they used to manage these challenges. This included understanding how they handled stress and which sources of support were most effective for them.

### 2.2. Participant Selection

To recruit participants, we contacted the international student office, various academic and honorary student organizations, cultural and religious groups, as well as the financial aid and scholarship office, requesting that they distribute the Qualtrics survey link to international students. At the beginning of the online survey, we collected demographic information (as shown in Table 1) and assessed their willingness to participate in an in-depth interview. Potential participants selected for in-depth interviews received email invitations. In total, 34 participants completed the in-depth interviews.

### 2.3. Interview Setting

We used Zoom (Zoom Video Communications, Inc., San Jose, CA, USA) to conduct online semi-structured in-depth interviews in compliance with local social distancing policies in the San Francisco Bay Area during the COVID-19 pandemic. A total of 34 interviews were conducted online via Zoom, with one participant and one interviewer. The interviews took place between 30 April 2021 and 11 June 2021.

### 2.4. Data Collection and Instrument

All participants were scheduled for a single interview, with an average duration of 50 min (ranging from a minimum of 20 min to a maximum of 101 min). Each interview was audio recorded through Zoom. The audio recordings were initially transcribed using Zoom’s built-in transcription feature, after which researchers manually edited the transcriptions for accuracy. These transcriptions were accessible exclusively to the research team. As an incentive for participating in the in-depth interviews, all participants received a $20 gift card. We focused on nine questions related to the purpose of this study, as follows (Table 2):

### 2.5. Data Analysis

QSR NVivo software (QSR International, Pty, Ltd., Doncaster, Australia) was used to analyze the 34 in-depth interview verbatim transcripts. Using qualitative analysis, the research team reviewed the transcripts and developed a set of codes derived from each question using two approaches, deductive and inductive coding. Table 3 demonstrates a summary of themes, categories, and codes from this study. The categories and codes from Table 3 are organized from the highest to lowest rooting index and density. Categories more frequently described by participants within the sub-themes are listed first in the table. Likewise, codes are listed in the order of density within their respective categories.

Figure 1 illustrates the interrelationship between the categories and codes of challenges both before and during the COVID-19 pandemic. Solid black lines indicate the codes and categories for each sub-theme, while dotted blue lines connect codes that were frequently mentioned together in the in-depth interviews. Specifically, three categories were consistently identified as ongoing challenges: identity crisis, racism and discrimination, and financial struggle.

### 2.6. Ethical Considerations

This study received approval under an exempt review category from San José State University (SJSU)’s Institutional Review Board (IRB). SJSU IRB granted a waiver of signed consent for both the in-depth interview and online survey. Consent notices were provided to participants at the start of the online survey and were subsequently emailed to participants before the in-depth interview. Permission for audio recording was obtained from all participants involved in the study. At the beginning of each interview, participants were informed that they had the option to use a nickname or pseudonym to ensure confidentiality, and they could also choose to turn off the video in Zoom to further protect their identity. To prevent identification, pseudonyms were used for each quote in the Results section, as well as in the demographic table (Table 1). An online random pseudonym generator was employed to create pseudonyms for the participants [29].

## 3. Results

Participants were asked about the challenges they have faced as international students in the U.S. and how they have managed to cope with those difficulties. Their experiences (i.e., challenges and coping strategies) were divided into the following three domains: (1) experiences before COVID-19, (2) experiences during COVID-19, and (3) coping strategies before and during COVID-19.

### 3.1. Experiences before COVID-19

International students encountered a variety of difficulties when pursuing higher education in the U.S. before the COVID-19 pandemic. Additionally, nearly all participants also shared some positive experiences related to being on campus and having face-to-face interactions. Participants’ experiences before COVID-19 were extracted and analyzed below.

#### 3.1.1. Challenges

##### Identity Crisis

Loss of identity was a significant challenge. Sean described the adjustment process in the U.S. as a “bleaching process.” He had to move on from his past life, letting go of his hobbies, friends, places, and food. Sean felt he had lost a big portion of his life. As he moved to the U.S., he had to focus on the present, which meant working and achieving his goals. He felt he had lost a sense of who he was and suffered from an identity crisis after leaving everything behind from his home country:
Because moving here, it was very much like a bleaching process. It’s like you pack your entire life into three suitcases, leave everything behind, and you just plopped here, and now you have to work and you have to achieve what you came here for. Your hobbies you used to have, the friends you used to talk to, the places, the food that you used to eat, the places I used to know, all that it’s just part of the past now, and you’re kind of, almost like completely bleached as a person. (Sean, age 25, male, Lebanon)

In addition to the overwhelming adjustment process, Stephen mentioned feeling the pressure to grow up and become independent as he spent time in the U.S. without his family. He noted that education in the U.S. is more costly for international students than for in-state students. The financial burden of affording university added more pressure to complete his education. Stephen expressed the weight of being responsible for his own life. As he began a new journey in the U.S. without his family, he could no longer rely on his parents to take care of things for him and could no longer depend on the “cushy environment”:
Education over here, COVID or not, is expensive and therefore the weight on the responsibility of completing the education would be heavier compared to being an in-state student …. That’s one of the things that no one really understands, and you’re going to have to take care of your living costs and stuff. You can only work on campus and nowhere else and that’s probably the only source of income that you can get apart from your parents. So the gist of it is being independent and responsible for every facet of your own life, whereas before … you’re living in a cushy environment for your parents [to] take care of everything, but that changed. (Stephen, age 24, male, Malaysia)

##### Limited Resources

International students did not have access to several resources, making their study abroad experience more difficult. Those who were able to find campus resources often encountered difficulties accessing them. Susan, for example, had an experience bouncing between different colleges within the university. She spent time searching for places on campus where she could access resources. During her first year at the university, she was afraid of falling behind academically and sought academic advising and tutoring. According to Susan, she had to make an effort to “dig through” the available resources because some websites were hard to navigate and the process was overly complicated:
Just going to the library all the time, but that was useful because I end up with a bunch of resources that you usually wouldn’t know if you just sit in a class and go home. Yeah, I think there’s a lot of resources that you have to dig through partly because like [in] the U.S., school website is hard to navigate. You have to Google everything, [including] academic advising. (Susan, age 20, female, Vietnam)

Some international students went to a community college and transferred to a university to complete their higher education. Upon moving into a different school, they faced challenges with transferring course credits. Robert expressed that he did not receive sufficient information on which classes he needed to take and had no choice but to create his own class mapping. He emphasized that international students might take unnecessary classes, which can result in wasted time and money. Robert voiced the need for universities to provide better guidance on the transfer process for international students:
I need information about school because I took courses in community college before transferring to a university. It’s kind of bad for me … because I don’t get enough credit information about transferring. Because you randomly took the classes [at the university] that you don’t need to, [which would] waste time and money. (Robert, age 32, male, Thailand)

##### Racism and Discrimination

Hate speech is a form of racism that verbally attacks a person or a group based on their race, ethnicity, nationality, color, religion, gender, or other identity factors [30]. International students, coming from outside the U.S., often faced racism and discrimination. Evelyn shared her experience of being discriminated against while walking in the city of Santa Cruz. A homeless person told her that she should not be in the U.S. and she was accused of stealing jobs. The homeless assumed Evelyn was from a foreign country because she looked Asian and did not speak fluent English. Evelyn shared her thoughts on how hate speech may negatively impact international students:
I was walking downtown Santa Cruz one day and then this homeless person was like “Stop coming to the United States and stealing our jobs.” Just because I look like Asian and my English wasn’t fluent, … he immediately thought that I came here from [a] foreign country. He said those things, I wasn’t really offended but I was like “These things happen to those innocent people who just want to come here and study”. To realize that, it’s not just American dream and then happy all the time. That was a challenge for me. (Evelyn, age 23, female, Japan)

##### Language Barrier

Language proficiency affects one’s ability to learn subjects [31]. Some international students with language barriers experienced difficulty learning at school. As a student in the U.S., Evelyn faced the challenges understanding academic terms. She was surprised that the words people use in daily conversations were so different from the words people would use to write scholarly essays. Evelyn took an anatomy class and struggled with the scientific terminology used in class. She mentioned that she knew basic English when she was in her home country, Japan. Despite this basic knowledge, she did not have the opportunity to practice speaking English. She expressed that communicating in English was challenging. With her shy personality, she hesitated to ask questions whenever she had a hard time in class:
Learning academic terms was really hard for me …. I didn’t know the academic term as well as [the] speaking term, like people use [a] different language from what they use in essay[s], right? So it was really hard and then one time I was taking anatomy …. In anatomy, you learn terms like medial and lateral. And I was like, “[Are] our nipples medial or lateral to shoulder?” I was like, “What are nipples?” …. I straight up didn’t know. But I was still shy to ask those questions [be]cause I thought that everybody was expect[ing] to know those terms. (Evelyn, age 23, female, Japan)

Some international students had the opportunity to learn English prior to coming to the U.S. However, the English they learned in their home countries was formal and not suitable for daily conversations. Nancy expressed the challenges of understanding slang and informal language. She knew some English before leaving Vietnam but was not familiar with how people in the U.S. use words in everyday communication. Nancy also mentioned that she was reluctant to tell people when she did not understand something because she felt embarrassed:
Another challenge that I had is the language, the language that people use here is very different from the English that I was taught. People use a lot of slang and a lot of informal words that I have no idea what they’re talking about …. I feel kind of ashamed if I tell people that I don’t understand what they’re saying. (Nancy, age 23, female, Vietnam)

Language barriers limit one’s ability to communicate with others [32]. For international students, low proficiency in English hinders their ability to voice their opinions. Edward described trying to learn English as “a survival moment”. He mentioned having difficulty expressing himself and that, being typically shy, he became hesitant to communicate in English:
I think the biggest challenge for me was trying to learn English. Just becoming fluent in English because I’m a shy person, and so I didn’t want to talk if I didn’t know what exactly I was talking about. I was trying to learn the language first so that I can express myself fully. I mean even now that [I’m] a little bit more fluent, I still can’t fully express myself sometimes. (Edward, age 23, male, Philippines)

##### Academic Challenges

Some international students faced academic challenges, such as a heavy workload. Denise spoke about the course workload in her hometown in India compared to that in the U.S. She mentioned that, back home, she would only have to attend class once or twice a week. In contrast, classes in the U.S. can last for hours each session. Additionally, students in the U.S. are assigned homework and readings that take up a significant amount of time. Denise also described one of her classes where the professor assigned readings and gave closed-book pop quizzes in class. There was also a debate session in the class in which students had to express their opinions on various topics. Denise, who had a language barrier, said she would spend a long time thinking about what she wanted to say:
The workload is very heavy. In my hometown, you just have to come to a class one or two times a week. In the States, you attend a class [for] like three hours. The homework and assignments and a lot of reading probably have to take three times to do it …. For example, one of my class[es], you have to read all the textbook beforehand. So when you come to class, the professor just give[s] you a pop quiz, five questions and then you close your textbook …. We [then] get into the debate session, you have to tell your own opinion, but I was so shy, and my English [wa]s not good enough, so I ha[d] a lot of thinking, but I [could] not say a word in front of them. (Denise, age 30, female, Taiwan)

Being abroad without family can lead to stress and loneliness [11]. Nicole shared her unique experience, as she is the first generation in her family to receive higher education and obtain a degree from a university. As the only member of her family with the opportunity to attend college, she felt like there was no one she could talk to whenever she had a hard time. She expressed how glad her family was to see her attending college. However, she always worried about her family knowing she was struggling. Nicole believed that many international students were concerned about disappointing their families:
Something that was really hard for me was when I was really stressed. I didn’t really feel like there were many people I could tell because no one in my family had gone to college before. They were all so happy that I was going to college, and I was doing things. That was good, but then it’s kind of stressful because when I didn’t do well, I would always worry that I [would] disappoint them, and I think a lot of international students have the same thing. (Nicole, age 25, female, South Africa)

##### Financial Struggle

In the U.S., only citizens and eligible non-citizens are eligible for federal student aid [33]. Therefore, many international students would take out loans to pay for their school tuition. Ernest expressed that he took out a loan, which placed a significant financial burden on him. He was concerned about paying off his student loans due to their high interest rate. As a result, even after graduation, international students would still have to continue paying off their loans:
The main challenge every international student would face is financial stability. Many of them would have taken loans and there is always a burden or oppression of satisfying God for completing the loan as soon as possible to get a good job, and then they pay off the loan because the interest rate is quite high in India, and that has to be paid very quickly or else it becomes quite a big burden. (Ernest, age 26, male, India)

International students faced limited job opportunities in the U.S. due to their legal status. Lois compared the work opportunities in the U.S. with those in Canada. International students in Canada can work outside the university, whereas in the U.S., they are restricted to on-campus jobs [34,35]. Lois also mentioned that on-campus student jobs are often low-paying and that there are not enough vacancies, resulting in a low chance of securing such a position. Lois said that, overall, there was not much financial help for international students:
In Canada, as far as I know, students can work out[side] of the university for some time and that helps them financially. But in America, we are not allowed to work while we are studying. We can work at [the] university and the payment is too low. In some universities, the positions get full very soon …. If in the mid-semester, you go and just apply for a job, your chances [are] really low to get that job. I think that’s one of the problems studying in America as an international student. There are not much financial help. (Lois, age 25, female, Iran)

When coming to the U.S., international students need to deal with currency exchange. Stephen started by talking about the challenges students face in finding a job. He pointed out that even before COVID-19, there were few on-campus jobs, and working in person was still challenging. At the time of the interview, Stephen mentioned that he was still in the job-hunting phase. He then expressed how the conversion rate from his home country’s currency to U.S. dollars could be intimidating. Some expenses, such as food, are much higher in the U.S. compared to his home country, Malaysia. He concluded that having a job in the U.S. and receiving money in U.S. dollars can help alleviate the currency exchange problem:
I think the challenges, personally, it’s getting a job. There’s not a lot of on campus jobs to begin with, even before COVID, so it’s hard for me to land a job on campus. I’m still applying right now, hopefully, for fall to get more opportunities but yeah, if you’re not working here, the conversion rate of your original currency to U.S. dollars can be a little daunting. Because if you want to spend money on food here it’s pretty much four times the amount you normally spend in your own country and so being able to work here would allow you to bypass that currency exchange rate. (Stephen, age 24, male, Malaysia)

While Stephen talked about the cost of food in the U.S. being higher than in Malaysia, Cheryl expressed the burden of converting Indian rupees to U.S. dollars to pay for college. She mentioned that she had to use two to three months’ worth of her salary to cover her college application fee. She noted that everyone is charged the same admission fee in the U.S., referring to it as an “an umbrella fee”. Cheryl also expressed the opinion that colleges should adjust admission fees based on students’ income:
Specifically for students from countries like India, where they just took a conversion of Indian money into U.S. dollars, you will end up paying a lot of money …. For my application fees, I had paid more than two or three months of my salary just [to] apply to colleges …. I think this is something that colleges should also look into your admission fee. It shouldn’t just be an umbrella fee for everybody, because clearly everybody’s not from the same background. It could be a little bit different depending on what country you’re from, or how much you’re earning. (Cheryl, age 25, female, India)

##### Culture Shock

Adapting to a new country and environment, Sean expressed being in an “extremely hyper-nervous state” during his first three months in the U.S. He had to adjust to the laws and regulations of the country. Sean specifically mentioned the jaywalking culture in the U.S., talking about how he was terrified every time he tried to cross the street. He felt unsure about the rules in the U.S. society because he did not know how things worked, which led to stress:
You come from very different countries …. Literally how you cross the street is completely different and so during the first three months I’d say you’re in this extremely hyper-nervous state or just you’re trying to learn the rules and norms …. That was something that I was absolutely terrified about, but for people here, they just do it regularly …. You’re very just unsure because you don’t know the rules of society here. You don’t know how things work and it’s kind of a bit stressful. (Sean, age 25, male, Lebanon)

Another culture shock that international students may face in the U.S. is individualism. The individualistic culture encourages people to create their own lifestyle and value self-reliance above all else. Shirley expressed that she came from a country where people are close to each other and rely on one another like family. She was shocked to find that in the U.S., people value personal space and are more focused on looking out for themselves. This made Shirley reflect on the things she does for the people around her:
I was born and raised in a country where … everyone knows everyone, and everyone was very nice and kind to each other and they act as if they’re one big family. And when I came here, everyone [wa]s just looking out for themselves and everyone really ha[s] this big idea of personal space. That kept me re-evaluating every action I d[id] with the people around me. (Shirley, age 18, female, Iraq/UAE)

#### 3.1.2. Positive Experiences

##### Offline School

Besides the challenges, there were some positive aspects of being in the U.S. As newcomers, international students sought to make the most out of their college experience. Stephen shared his daily routine on campus, which involved arriving in the morning and staying until the evening. He mentioned that he enjoyed meeting people during his time at school and felt like he “basically lived on campus compared to actually living at home”:
Before [COVID-19], I would go to school in the morning and I would just stay there until the evening even though I didn’t have class [be]cause there was always something to do. There were people to meet, and you can study in the library, which I kind of liked and you could get involved too …. I would just basically live on campus compared to actually living at home. (Stephen, age 24, male, Malaysia)

In addition to having a pleasant time on campus, international students valued the opportunity for in-person learning. Ben shared his positive experiences at the university, emphasizing that he enjoyed being physically present in his classes. He also liked the Socratic seminar type of learning. However, he mentioned that he did not like video lectures because there was no interaction. Ben concluded that his higher education experience was great before the pandemic:
It’s pretty fun [be]cause [university name] is a pretty big campus and I’ve been enjoying the seminars. I’m the guy who likes to be in class because I like the Socratic seminar type of learning. I really hate video lectures and with no interaction whatsoever. So it was really going well before the pandemic. (Ben, age 24, male, Myanmar)

As international students had the chance to experience offline school, they were able to meet new people and build friendships. Kelly expressed that she was able to socialize and communicate with people. She met friends and professors who came to the U.S. as an immigrant. Through her conversations with them, she gained a better understanding that one can overcome challenges through hard work. Kelly claimed that she had a great experience before COVID-19, as it allowed her to practice her communication skills:
I made new friends. [I was] more open-minded and more socialize[d] to talk to everyone and I learned a lot [from] my friends and my professor who came before me, like immigrants, and understand that if we try hard, we can overcome challenges. Before COVID, my experience [was] really nice and interesting because I had a chance to practice communication skills. (Kelly, age 23, female, Vietnam)

### 3.2. Experiences during COVID-19

Due to the COVID-19 pandemic, students were compelled to stay at home and continue their education through online learning. International students voiced the challenges they faced, including those that arose during the pandemic and pre-existing issues that worsened due to these unprecedented times. Below, we gathered and analyzed participants’ experiences during COVID-19.

#### 3.2.1. Challenges

##### Escalated Identity Crisis

Identity crisis further escalated in the midst of the COVID-19 pandemic. Sean had previously expressed the challenges of losing his identity before the pandemic. Once again, he referred back to the term “bleaching process.” He mentioned that everything was put on hold during these unprecedented times and added that the self-identity crisis he had been facing persisted for an extended period:
When I said how it bleaches you, [the] chance to rebuild your hobbies, it put all of that on hold. It just completely stopped that and you were kind of stuck in that self-identity crisis for much longer than you would be with no idea what’s going on [and] why you’re feeling the things you are feeling. (Sean, age 25, male, Lebanon)

The COVID-19 pandemic also led to a pervasive and profound sense of uncertainty about the future. Lois expressed that international students, including herself, longed to visit their family. However, they could not leave the U.S. for fear they might not be able to return and continue their studies [20,25]. There was no certainty regarding when schools would reopen. Lois mentioned that this uncertainty left international students with “a big question mark”:
There are many international students that wanted to visit their family …. We knew that we [could] not go home [be]cause if we went home, we couldn’t get back to school and we didn’t know when the school w[ould] be open so everything was a big question mark and nobody could decide what to do. (Lois, age 25, female, Iran)

##### Higher Experience of Racism and Discrimination

The COVID-19 pandemic heightened racism and discrimination against Asian Americans. Raymond, who came from Vietnam, experienced discrimination as people believed that the Chinese were spreading the virus. He shared his sister’s experience during grocery shopping, where some Americans told her that masks were not allowed in the facility. His sister was also told to leave the facility if she was feeling unwell. Raymond stated that the discrimination negatively impacted his sister’s mental health:
Asian people like me sometimes feel discriminated because of COVID. Some people believe [the] Chinese spread it around the world. For Asian people like me, when going out to buy food, [we get] attention from other people. This is the story of my sister. [When] she [was] going out to [the] supermarket in Mississippi to buy groceries, American people came and asked us to take off the mask [saying], “You’re not allowed to wear [a] mask in supermarkets. If you are sick, just go back home” with [an] aggressive voice and she [(her sister)] called me [saying she was] feeling scared …. It took her months to get things settled. (Raymond, age 28, male, Vietnam)

In addition to racial exclusion, the COVID-19 pandemic exposed international students to feeling of inferiority. Linda mentioned that news and social media were the two platforms where people posted unpleasant comments about international students and their home countries. She expressed that seeing the unwelcoming comments led to negative emotions and that she constantly felt these negative feelings whenever she saw the news:
If you look at social media or [the] news, there are a lot of bad comments [about] international students and some negative comments [of] my country … This gives you some negative emotions … You have the feeling all the time [when] you look at a newspaper, watch the news [on] TV. (Linda, age 29, female, China)

##### Online Learning Challenges

International students reported facing online learning challenges during the pandemic. Sean shared his hard times dealing with the lack of separation between learning and his home environment. He defined COVID as “stagnation” and stated that the pandemic affected both his personal and professional life. Due to the stay-at-home mandate, Sean was forced to remain at home, where he dealt with hardships that reduced his ability to focus on his studies:
COVID was a stagnation. While it [(COVID-19)] goes outside the educational circle, it still affects your personal and professional life. If your personal life becomes this chaotic, it creeps into your education, your research work …. When you’re forced to go through [hardships] at home, [the] internal mess you have with your personal issues are brought into class and it did affect your ability to focus to study because you can’t go to [the] library anymore. (Sean, age 25, male, Lebanon)

Debra also shared her experience with online learning. She mentioned that she was not able to focus because she had all her classes and meetings on Zoom, leading to fatigue. She expressed as if she had not learn anything, as she spent most of her time going through materials but not as much on school assignments. Debra concluded that online learning was something she was not used to:
It’s definitely not better. I don’t know how I passed my classes because I cannot focus …. All these zoom classes give me fatigue and I have a lot of meetings too …. I’m basically just going through materials and not going that in-depth [with] my schoolwork, so I would say I’m not learning anything … Virtual learning, that’s definitely not what I’m getting used to. (Debra, age 20, female, Ukraine)

Not only did online learning lower one’s ability to focus on their studies, but learning at home during the pandemic also led to academic stress. Nicole expressed that all she wanted was for someone to ask her how she was doing. She stated that she hoped someone could ask her about how things were going for her. Nicole believed that checking in with someone is a sign of care. Since no one ever asked her, she felt as if no one cared:
You don’t want to do work [and] I don’t really interact with a lot of people. You just want someone to ask you, “How are you doing? I know you had an exam last week, how did that go?” To have someone [to] send you a message and say, “Hey, I know you were really stressed about that. How did it go?” But nobody does that. It doesn’t really feel like anyone cares. (Nicole, age 25, female, South Africa)

In addition to academic stress, international students experienced burnout due to online learning. Jonathan stated that he felt tired and burned out because he used his room both for sleeping and to studying. He shared that online learning had a negative effect on his mood and overall health:
I got burned out more easily [be]cause I slept in the same room [and] studied in the same room, so everything is so confined and I felt tired most of the time. I think that has affected my mood and [my] overall health a lot, which may have contributed to me burning out quicker. (Jonathan, age 22, male, Malaysia)

##### Heightened Financial Struggles

International students in the U.S. are only permitted to work on campus [35]. However, due to the COVID-19 pandemic, school campuses were forced to close [36]. Earl, who first enrolled at a university in the U.S. in the Spring of 2020, expressed that he had a hard time securing an on-campus job even in his first year. There were not many jobs available to begin with:
It was really difficult and challenging to get [an] on-campus job and there was no money [(income)]. The campus was closed [and] there were only a few jobs that were open so that made it really hard in the first year to get [an] on-campus job. (Earl, age 26, male, India)

There may have been some cases where people received a job offer right before the COVID-19 pandemic hit. Matt was fortunate to secure an on-campus opportunity, but as the pandemic began, the offer was rescinded. He lost his opportunity and became stressed, as he had hoped to use the money he earned to manage his finances. He then tried applying for TA positions and fortunately was able to receive three offers. Matt added that many other international students had to manage the process of losing a job and finding a new one:
I had an on-campus opportunity [but] because of this lockdown, the offer was revoked. They had to cut down many people because they don’t need [them] right now. That actually created a little bit of stress. I [thought] that I would get this opportunity to manage expenses, but all of a sudden, I don’t have it anymore. I tried reaching out to professors for TA positions and I got three positions, but not all of us [(international students)] could get [the opportunity] so some people had to manage. (Matt, age 29, male, India)

As schools transitioned to remote learning, students began to question the rationale behind paying the same amount of tuition and fees. Nancy expressed that international students were more burdened by the unchanged amount because they pay higher tuition and fees than other students to begin with. She mentioned that it was a waste of money, especially since international students had come all the way to the U.S. seeking in-person experience, networking, and meeting people. International students paid a large amount of tuition mostly just for online learning:
Students were very frustrated about having to pay the same amount of tuition even though we didn’t go to school and use all the stuff that we’re supposed to use. But for international students [it] was worse because we have to pay like twice or even three times more tuition and [that] is really unfair …. [It] is a waste of money because when we came to the US, [it] is not just what we learn but it’s the experience of going out and meeting people and making connections and see[ing] how things go here. (Nancy, age 25, female, Vietnam)

##### Legal Status Concerns

Maintaining a valid visa is essential for international students to continue their studies in the U.S. [37]. However, travel restrictions and lockdowns due to the pandemic made it challenging for international students to undergo the visa renewal process. Debra was perplexed as her visa was expiring. She expressed that the uncertainty of the pandemic added pressure to her concerns about legal status:
I couldn’t see the end of the pandemic and my visa [was] expiring so I’m like, “What’s going on?” The time just [went] on and I’m still not learning anything. (Debra, age 20, female, Ukraine)

In addition to visa expiration, international students faced the challenges of fulfilling the requirements to maintain their legal status. Nicole expressed that international students were stressed because they had to constantly keep track of their visa status. She mentioned having to align her major requirements and course timeline with her legal status:
It’s really stressful for international students because we always have to remember our visa [status]. The unit requirement [for] my major, some of the classes are only offered in certain semesters. So the senior project, for example, you can only start in the fall semester. So if I don’t start this fall, I have to postpone my graduation by another year. And [I] can’t start the senior project if [I] don’t pass the class …. You kind of have to stress about it. (Nicole, age 25, female, South Africa)

International students faced concerns about returning from visiting family in their home country [20,25]. Lois expressed her fear of being denied re-entry, as the global response to the pandemic led to travel restrictions, quarantine measures, and re-entry requirements. The rules for re-entry became complex, making it challenging for international students to navigate through them. Additionally, the fear of denied re-entry took a toll on international students’ mental health. Traveling during the pandemic was associated with stress and uncertainty:
I remember last summer, one day, I was checking my Twitter feed and there was a tweet about students who are starting here [(in the US)] fully online, they had to get back home. And for me, … I was crying because I didn’t know if I c[ould] get back [and] when the COVID [would be] over. I was like, what if I go home and I cannot get back and the schools reopen. I should get a visa again and everything will be like hell. It was really annoying. I was crying a lot. (Lois, age 25, female, Iran)

##### Lockdown Protocol

International students experienced isolation and loneliness during the pandemic, which had mental health implications [38,39]. Debra reported feeling anxious and depressed as she lived alone in the campus dorm, with no roommates. She lost motivation to attend class and became anxious about the future:
I’ve lived by myself the whole pandemic. I moved and I have no roommates in my dorms and I feel that’s going to make me really sad and isolated from people so sometimes I wouldn’t have motivation to even go to those classes. Just the anxiety of the future, … anxiety and depression. (Debra, age 20, female, Ukraine)

Lockdown protocols drastically limited social interactions [40]. For international students, who often relied on meeting friends and peers to combat homesickness, these restrictions were devastating. Carl expressed that there was no place he could go besides the grocery store. He lived by himself and had not made many friends yet. Carl felt depressed during the pandemic due to his little interaction with others:
I was just living by myself [with] sad human interaction. Right after the lockdown, I only went to [the] grocery. Personally, there was nowhere else to go. I was just living by myself [with] no human interaction. I didn’t make a ton of friends either. It was pretty much like a basic level of friends, so it wasn’t close friends. That was pretty much it during the whole lockdown. After COVID [was] pretty depressing. (Carl, age 23, male, Cambodia)

The pandemic led to travel restrictions and lockdowns, making it nearly impossible for international students to return home. The separation from their families added an emotional burden, intensifying feelings of loneliness and homesickness. Eric expressed that he missed his family but was fully aware that he would not be able to travel due to the pandemic. Eric faced the challenge of homesickness, having not seen his family for a couple of years:
I have not seen my family for two years now. I plan to stay during the summer because back in my home country, things are not going quite well. So that’s a challenge. I miss my family a lot and I know that the visa requirements, you need to obey. And because [of] the pandemic, I cannot travel. (Eric, age 20, male, Vietnam)

As lockdown protocols were implemented, international students struggled to maintain a balanced routine, eventually experiencing a loss of structure in their daily lives. Nicole expressed that she was “not in a good mind space” and shared her experiences of the changes to her day-to-day structure due to the COVID-19 pandemic. Outdoor restrictions and the shift to online learning affected her mental health, leading to a loss of motivation and productivity:
I don’t think it’s been really good academically because you’re not in a good mind space. Some days, it’s really hard to wake up and you just want to sleep. You don’t want to do work, but then you have to. And before, I wasn’t like that. (Nicole, age 25, female, South Africa)

#### 3.2.2. Positive Experiences

##### Convenience of Online Format

There were university campus closures due to the COVID-19 pandemic. Although most participants favored offline learning and face-to-face interactions, a few international students appreciated the convenience of the online format. Bruce expressed that he preferred online learning over in-person classes. As a student who relied on public transportation, he saved time by having classes online. He no longer had to wake up early to catch the bus and spend hours commuting to campus:
My experience [with] online learning is so much better. To be honest, right now, I prefer online learning rather than in person. I don’t have to go outside. Because I don’t drive, I take the bus and other public transportation so it can be a little pain in the butt to get up early and you have to sit on a bus for an hour or two to get to school and back. (Bruce, age 21, male, Iran)

In addition to online learning, some classes included lecture recordings, allowing students to review the materials as many times as they needed. Lecture recordings can be immensely helpful for international students with language barriers. Denise mentioned that she had a teacher who spoke relatively faster than other teachers. She stated that even native English speakers were having a hard time catching up with the pace. Denise found that being able to watch lecture recordings repeatedly helped her get caught up with the materials:
We cannot imagine if there’s no recording, how can we survive …. Because, for example, we have the statistics class and the teacher speaks so fast. [The teacher is] from India and she taught well, but she speaks so fast. So even native students, they said that the professor speaks three times [the] normal speed. So we will have to watch the recording. That’s the good thing. (Denise, age 30, female, Taiwan)

### 3.3. Coping Strategies

Participants were asked how they managed the challenges of being an international student in the U.S. before and during COVID-19, and how they managed stress regardless of the pandemic. Although coping strategies before and during COVID-19 overlapped, there was an increased frequency of several coping mechanisms due to the pandemic. In addition, many participants shared their desired support systems, as well as how their family and closest individuals contributed to their support systems. The participants’ coping strategies have been grouped and analyzed below.

#### 3.3.1. Managing Challenges as an International Student before COVID-19

##### Physical Activity

Physical activity, such as exercising or going to the gym, was one of the major coping strategies for many participants. Engaging in physical exercise contributed to better mental well-being and improved their quality of life. Sean mentioned that going out for a run or jogging gave him a sense of accomplishment in his daily routine and helped him clear his mind from the struggles of living in the U.S.:
Exercise, going out for running and jogging, that was a major head clearer …. Because when everything … is flatlined and … everything’s the same indoors, exercise gives you a bit of adrenaline, a bit of dopamine. It gives your day a peak, that little achievement in your day. (Sean, age 25, male, Lebanon)

In addition to exercising, participants mentioned that exploring and traveling to new locations helped them manage their challenges by providing a break from the stress of their daily lives. Whenever Debra faced the struggles of living in the U.S. as an international student, she found that exploring new places and being surrounded by nature brought excitement into her life:
I love traveling, I love exploring new places. I feel I’m the one who always finds new places when my American friends are already fed up with everything. So they like going to places with me because I’m always excited and so happy and screaming when I see the ocean. That’s been the greatest part. (Debra, age 20, female, Ukraine)

##### Social Exposure

Since many international students come to the U.S. without prior connections to the community, several participants mentioned that involvement in school clubs or campus events had helped them meet new friends and start building a community. Nicole stated that she enjoyed her life on campus and got along with her classmates and professors. In addition, she was able to meet more people through her on-campus job as a tutor:
Before COVID, I think it was really good. I spent a lot of time on campus because I took a lot of classes because I really liked learning. I was hanging around on campus. I liked campus. It felt like a safe place to be and my teachers, I got along with people …. I was also a tutor so that helped me connect with people somewhere too …. I spoke to more people, and I would see more people and we cared about how people were doing. (Nicole, age 25, female, South Africa)

Having an international student community helped international students overcome the challenges of living in the U.S. Since international students often share similar struggles, being able to share their experiences with each other’s problems provided a sense of solidarity and made the participants feel less alone:
I have a lot of international student friends and [the] challenges they were facing, I am facing as well so we share experience[s], especially during the lockdown …. [It’s] kind of reassuring to know that you’re not the only one, you’re not alone. (Carl, age 23, male, Cambodia)

##### Self-Improvement

Hobbies such as painting, playing an instrument, and taking photos were a common coping mechanism for many of the participants. Engaging in activities they enjoyed helped students take their minds off life stressors and focus on improving their mental health. Nancy shared that one of her hobbies, planting, had helped her reduce stress:
I love plants. I have a lot of plants in my house. I’m a crazy plant person and having plants around actually keeps me happy. Plants [are] actually the reason that I feel a lot less stress. (Nancy, age 25, female, Vietnam)

Adjusting negative mindsets was a common method of self-improvement. When international students first arrived in the U.S., many struggled to adapt to the new culture and environment. They experienced additional stress from the pressure to blend in with the community. Edward claimed that in order to cope with the cultural differences, he tried to find his “true self” in the new culture rather than worrying about the differences:
We moved around a lot when I was young so blending into that culture is a necessity for me. My coping strategy when I first got here was to mimic as much as I could so that I don’t look different from the others. That was the main thing, and as I got better at that, I started working on myself, which is finding my true self in the culture that I’m living in. (Edward, age 23, male, the Philippines)

In addition, Evelyn mentioned that instead of stressing over the challenges of adjusting to the new environment, she changed her mindset to avoid being bothered by cultural differences or language barriers and learned to let go of negative thoughts:
I came to a point where I don’t care about mindset. Before, I couldn’t even talk to American people because I was so afraid of making mistakes in English but I was like, “It’s okay.” In my mind, they make mistakes too so that reduces a lot of stress. (Evelyn, age 23, female, Japan)

Effective time management was a common strategy for improving their class performances. Many struggled to adapt to the differences in academic curriculum and initially faced difficulties keeping up with their college courses. Linda mentioned that she struggled to maintain a good grade at the beginning of the semester. However, she was able to gradually improve by establishing to a healthy study schedule and finding her own study methods:
I got a very bad grade on my first midterm. It’s really bad. I never got a grade so bad. I was so worried because I [could] not fail …. I just got a better plan to study harder. Fortunately, I got a better grade next time …. I don’t know how others do it, but this is how I did it. (Linda, age 29, female, China)

##### Mental Self-Care

Several participants mentioned that meditation improved their overall mental health. As international students struggled to adjust to the new college life in the U.S. and learn to become independent, they experienced difficulties finding methods to address their worries and problems. According to Sean, meditation was like a “portal” that helped him control his emotions and organize his thoughts. Having a moment to acknowledge the hardships he was facing allowed him to reflect on his life and develop solutions to his struggles:
It was meditation, most importantly breathing exercises. I found that it’s a very effective way of controlling your own thoughts. Learning how to control your thoughts, learning how to direct your thoughts, what to focus on because when your life is a bit of chaos …. You end up in a situation where you haven’t had someone to talk to you. You are just left with your thoughts so meditation was kind of a portal … to calming down and working towards [a] solution. (Sean, age 25, male, Lebanon)

#### 3.3.2. Changes in Managing Challenges as an International Student Due to COVID-19

##### Increased Frequency of Online Engagement/Communication

Maintaining communication with friends via online platforms was helpful for international students to stay connected with their community. As students faced restrictions on in-person interactions and social isolation due to pandemic lockdown protocols, many participants turned to online communication such as group chats and video calls. Ben stated that he had many video calls with friends from the two colleges he attended in the U.S. and that staying in touch helped him maintain his English skills:
During the lockdown, since you can’t go out, thank God for group chats and Zoom and video calls. [I had] a group chat with some poli sci [(political science)] [friends] from the same majors and we talk a lot, and we have game nights and movie nights. (Ben, age 26, male, Myanmar)

Participants mentioned that online communication helped them stay connected with people from their home country. Due to the international travel restrictions, many students were unable to return to their home countries. In addition to adjusting to the changes in living in the U.S. due to COVID-19, spending more time away from their family and friends added extra stress to international students. Patrick used online platforms such as Facetime and WhatsApp to communicate with his family in India:
You obviously miss the home food and thanks to the technology, for the stress, during the morning hours [I] speak to [the] home country people. I just use tools like Facetime [and] WhatsApp so I can connect with them and talk with them. (Patrick, age 23, male, India)

##### Elevated Motivation for Self-Improvement

Students spent most of their time indoors during the pandemic lockdown, with limited opportunities to socialize and participate in outdoor activities. Many participants used this time for self-improvement. Learning a new language was a common coping mechanism, as it provided a sense of motivation to acquire new skills. Several students reported having increased motivation to learn a new language during COVID-19. Alice enjoyed learning two new languages:
Since the lockdown, [I] was basically [at] home, but I tried to do something fun to take care of myself as well. I started to read and study seriously about other foreign languages. Currently, I started to learn Chinese and Korean. (Alice, age 22, female, Japan)

As international students spent spend more time away from their home country, many began to miss their home cultures. Learning to cook allowed them to stay connected with their roots. Edward mentioned that making food was his way of coping with life’s struggles. Cooking reminded him of his homeland and became a source of comfort during the COVID-19 pandemic:
I also like to cook …. Cooking really helped because I can still make the foods that I used to eat when I was in Japan or in Saudi Arabia or in the Philippines here in the U.S. Cooking is a really big part of my coping strategy. (Edward, age 23, male, the Philippines)

#### 3.3.3. On-Going Support Systems as an International Student Regardless of COVID-19

##### Campus Resources and Job Opportunities

One of the support systems participants relied on was the on-campus resources, especially the research assistant opportunities. Working with professors provided them with a sense of being part of the college community and helped them adjust to the new campus environment. Sean mentioned that he enjoyed working, as it gave him a sense of purpose by being involved in the projects:
I’m working as a research assistant. That helped me have some kind of tangible work …. Working with a project that had goals, rather than studying and having [an] exam …. [It gives] you a sense of purpose during that time. (Sean, age 25, male, Lebanon)

Department counselors and course professors have been a support system for several participants since moving to the U.S. Many international students experienced unease during their college life in the U.S. When first moving to the U.S., participants often had difficulties adjusting to the new educational system and finding campus resources due to their language barriers before COVID-19. Ben claimed that his professors had helped him throughout his college career. Especially during the COVID-19 pandemic, when he was socially and emotionally isolated from his community, his professors frequently stayed in contact and showed concerns about his mental health. During the interviews, multiple participants expressed their gratitude towards their college professors:
Some professors have been really amazing because since this pandemic, there has [been] a lot of problems …. Sometimes I just shut everything out, I mute all my social media … and then some of my professors know what kind of student I’m like. I’m the loud, talkative guy, and if they don’t hear me for a couple [of] days, a lot of professors have checked in …. So I’ve been really appreciating these people. (Ben, age 24, male, Myanmar)

The international student advising services were another common academic support system among the participants. Unlike the general campus counseling services, international advising services focus on the challenges specific to international students, including academic advising, career counseling, and immigration assistance. Lois mentioned that one of the reasons she chose her college was the guidance and support by her advisor:
I have a perfect advisor. She’s amazing and I will never forget her. She helped me and understood me a lot and I really enjoy working with her. I’ve been lucky to find her and actually, she was the reason that I applied for [university name] …. I’m really happy. (Lois, age 25, Female, Iran)

##### Family/Relatives and Friends

Many participants mentioned that family support was a strong support system during their time in the U.S. Although students struggled with financial difficulties, many parents greatly contributed to their living expenses by paying for their tuition and dorm fees. Ernest’s father had been supportive of his education in the U.S., and his journey through undergraduate had been smooth:
Coming out to the financial support, my dad was very supportive with my education [with] my plans of doing [a] master’s [degree] and none of them became a hurdle to me. (Ernest, age 26, male, India)

In addition to family support, many of them received emotional support from their relatives. Several participants considered their uncles and aunts to be like their second parents when they were not able to receive direct support from their families back home. Alice shared that her cousin’s family had been one of her greatest support systems in the U.S. When faced with a challenge, she would reach out to her relatives, and her uncle would frequently keep in touch with her. She also described her relationship with her cousin, who provided emotional support by listening and offering advice on her life problems:
Definitely, my cousin’s family [has been supportive]. My uncle calls me [at] a very random time but he’s always checking in with me like, “Hey, [how] are we doing?” That helps me and also, I met my cousin at the end of February, because she planned to meet her friend from high school in San Jose, so she invited me since I ha[d]n’t met [her] and I wanted to catch up with her too. These are the support, like family aspect.(Alice, age 22, female, Japan)

Furthermore, several participants mentioned that their relatives had supported them academically. Many of them were the first in the families to study abroad or graduate from college in the U.S. While their parents had been financially and emotionally supportive of their study abroad, their relatives living in the U.S. provided strong educational support. Betty shared that she relied on her relatives for academic advice because her parents were not able to understand her educational struggles as well as her aunts and uncles did:
I shared these [(school-related problems)] details … with my uncle and aunt. I used to go back [on] the weekends in the fall semester because they lived in [city name], … so I was able to tell them because they’ve both studied in [an] American university. They have more understanding of what I’m talking about so they were able to understand more than my parents. (Betty, age 19, female, Singapore)

Another common support system for international students was their friends. Regardless of the COVID-19 pandemic, college students often engaged in several online group chats with members of a club or with other students taking the same classes to academically support each other and stay updated with the class discussions. Furthermore, by sharing similar cultural backgrounds with group members, participants could more easily join new communities and build social networks with both local and international students. Eric shared his experience of participating in a group chat for Asian Americans. Although there were no international students in the Discord channel, a web-based communication platform, he mentioned that he felt comfortable socializing with the local students:
In class, we often have [a] Discord channel, … a social media [for] students in the same class and then there’s a channel for Asian American. In school, … even though there are no international students, I find a lot of common things to talk about because they are Asian American …. If anything happens, I can talk to them, … and they support me. (Eric, age 20, male, Vietnam)

Having a strong academic support system was crucial for international students. One of the challenges of moving to the U.S. was adjusting to the new academic curriculum [41]. Students, including Ralph, struggled to keep up with their courses due to language barriers and a lack of familiarity with the U.S. educational system. As the majority of international students were away from their families, many relied on their friends as support systems to help improve their academic performances. Ralph mentioned how his friends in the study groups guided him on how to pursue his future career:
I have a study circle. I talked a lot with friends here in [university name]. Once in a week, I actually interact with them and get to know what they’re studying and how to go over the rest of the semester or whatever we [are] actually pursuing. (Ralph, age 24, male, India)

#### 3.3.4. Desired Support Systems as an International Student

Participants were asked about the types of support systems they believe would be helpful for international students to be successful in living in the U.S. The responses have been categorized and listed below.

##### Financial Aid for Tuition Cost

Several participants mentioned their financial struggles. Despite receiving financial support from their parents for living and educational costs, students still faced difficulties managing their budgets. Debra mentioned that she hoped colleges would provide international students with financial aid and resources to help them find part-time or full-time job opportunities. She felt that her university fell short in informing new students about the available campus resources:
Financial help would have been nice …. I expect more from school because we pay much more [for] education costs. I feel our school lacks their promotion of the resources … [and] just connecting students to the available opportunities …. I feel like they [are] just [like], “Okay, you arrived here. You go figure out by yourself how Americans live.” (Debra, age 20, female, Ukraine)

##### Approachable Campus Mental Health Resources

Having more approachable campus mental health resources was another common support system mentioned by the participants. Betty shared her experiences with accessing mental health resources on her college campus. During her international student orientation, she was introduced to several college resources and noticed that some students, including her friends, felt uncomfortable reaching out to the counselors or advisors on campus. Additionally, Betty felt there was a lack of emphasis on the importance of mental health awareness. For students who struggle to seek support, she wished there was a more approachable campus resource available for international students to access mental health programs:
I know there are resources and there are places to ask for help, and this was told to us in orientation. They told us multiple times like, “Please don’t feel afraid. You can reach out to us, there’s no judgment.” And I feel like there are plenty [of] resources. But I feel [there are] students who don’t want to reach out [or ] … approach somebody …. I feel they wouldn’t share contact [with] somebody from the university for mental health …. I feel like overall, mental health has not [been] given as much importance or not talked about as much. (Betty, age 19, female, Singapore)

##### Welcoming Campus Programs/Clubs for International Students

Several participants felt the lack of welcoming campus programs for international students. Students struggled to adjust to the new academic environment and often relied on support systems to navigate through their first few years as international students. Nicole shared her in-depth experiences with her college orientation and advising program as a newly enrolled student. During her orientation, she struggled to register for classes and felt neglected by the orientation leaders as they were not supportive in helping her find her courses. In addition, she shared her negative experiences with the academic advisor from whom she did not receive the support she needed. She shared that she felt blamed for asking the advisor questions and felt that the advisors did not care about the students. She mentioned that having an advising program that genuinely provides international students with resources will help support their academic success:
It was my first semester at [university name], and I was really confused because I did the online orientation and it wasn’t super good. Even after the orientation, I didn’t know how to register for classes. Then I made an appointment and I was talking to her [(academic advisor)], and she was kind of rude and anytime I asked her something …. We have questions and things are confusing and they shouldn’t make us feel bad for having those questions. They should be helping us. They should be telling us, “We’re okay, we can do this,” standing up for us advocating. But sometimes it feels they’re just doing it because it’s their job, they don’t really care. So I feel for international students, they really need to get good advisors who actually want to help people and who want to teach them how to navigate the system.(Nicole, age 25, female, South Africa)

##### Transportation Resources

The lack of transportation resources was another challenge international students. Several participants mentioned that, since they did not have access to a car, their only mode of transportation was walking or taking public transportation, such as local buses. According to Shirley, campus resources that provide information about off-campus transportation would give international students the opportunity to explore locations beyond the campus:
I remember calling sick once, and I never went anywhere outside the campus place so I didn’t know where the hospital [was] or resources [to] drive me there or [the] bus stations [to] take. Even going to school by bus [was] a challenge to me. I got lost multiple times and I wish there were someone telling me go to this route or this route. (Shirley, age 18, female, Iraq)

##### Paid Research Opportunities

International students felt that having more paid research opportunities would support them financially and academically. Many participants experienced financial struggles, and having a paid on-campus research opportunity provided extra income. Furthermore, students could acquire new skills outside of their classes through the experiences of being a research assistant. Lois shared her involvement in on-campus job opportunities and mentioned that having more accessible research assistant positions might help students feel more motivated to study and increase their sense of independence:
I think better research positions, having more research positions in [the] university. Paid research positions make them more motivated, to study better and feel more independent. (Lois, age 25, female, Iran)

##### U.S. Federal Information (Taxes)

Sean shared his unique experiences with wanting to have more access to U.S. federal information in college, specifically related to taxes. He mentioned that filing taxes or tax returns in the U.S. is different from the system in his home country, and navigating the process was stressful for him. Therefore, he wished his college had provided international students with information related to U.S. federal tax laws:
I wish during orientation, they ha[d] talked to us about taxes. That one is very stressful to figure out on your own because doing taxes wrong here is like a big federal crime so you’re absolutely terrified about it. I wish the school did a better job during orientation to talk about this …. That was something missing [and] I think that’s very important to cover. (Sean, age 25, male, Lebanon)

#### 3.3.5. Closest Individual for Mental Support

##### Friends

Many participants mentioned their friends as their closest source of mental support. Alice and others shared how their friends provided emotional support by empathizing with the struggles of being an international student. When adjusting to the new environment and coping with the challenges related to language barrier and cultural differences, students relied heavily on their friends’ support. Alice was able to share similar challenges and overcome struggles together with other international students. Furthermore, she had local friends who offered new perspectives and helped her adjust to college life in the U.S.:
Sometimes, [I] reach out to my other international friends who’ve been here longer than me to [ask] advice. Also, my local friends here understand me and [the] kinds of challenges I have as an international student. They bring up their local perspective to see things [in] more than one way and help me to understand. (Alice, age 22, female, Japan)

Participants often struggled to find support systems that could closely help guide them through their college life and career. Earl mentioned that his closest friend had been one of his greatest mental supporters. When he had questions regarding his courses or when he needed help practicing for internship interviews, his friend always gave him advice and resources to help him improve:
I have one friend, [who is a] student as well …. He is [a] senior to me in [university name] …. When I have questions about [the] courses that I need to pick or the internship that I’m applying to, [he] helps [with] the interviews. (Earl, age 26, male, India)

##### Family

Participants considered their family members as their closest individuals for mental support. Among those interviewed, Evelyn mentioned that her parents were supportive of her decision to study abroad, which motivated her to work hard in the U.S. Although her parents were initially reluctant, they have been supportive ever since she moved to the U.S.:
When I told them [(parents)] that I wanted to go to [the] United States, even though it took a while for them to understand why I wanted to come here and why I wanted to do it, by explaining it, they understood and were supporting me since then. (Evelyn, age 23, female, Japan)

Another type of family support that international students experienced was related to their educational and future aspirations. Participants mentioned that having their parents support their educational and future goals motivated them to work harder in the U.S. William shared how his family supported his decision to study abroad and looked forward to his future success:
[I] talk with them [(parents)] … and they will help me to learn what I [should] do in this situation, just giving me some advice, and this is very helpful. They always say … “You are still young, you can have lots of problems. You have a great future waiting for [you]”. (William, age 25, male, China)

As international students, many participants mentioned the importance of receiving unconditional emotional support from their families. Moving to the U.S. alone often led to experiences of homesickness and loneliness. Especially during COVID-19, when making new friends and connecting with the college community became challenging, spending time with family became a crucial coping mechanisms. Lois mentioned that her family’s support helped her become emotionally resilient while living abroad alone:
Having a supportive family is something that is really needed. There must be someone who says “Okay, I care about you. Don’t worry, I’ll be here for you and everything will be fine”. Fortunately, I have this situation. (Lois, age 25, female, Iran)

##### Partner

Another common support system that participants often mentioned was their partners. Shirley mentioned that her boyfriend provided her with emotional support. Whenever she needed someone to talk to or seek advice from, her boyfriend was always there to listen and support her. Although she felt he was not aways able to offer practical solutions, he served as a strong source of mental support:
My boyfriend, he’s not an international student. He lives here … [so] my boyfriend is always there. Whatever I have in mind, I always go to him. He cannot offer much help, but he can offer emotional help. He never rejected me when I came to him and asked him for help or advice. (Shirley, age 18, female, Iraq)

In addition to the emotional support students receive from their partners, Carl mentioned that sharing future goals with his girlfriend helped motivate him to work in the U.S. He plans to eventually obtain a permanent residency and build a life in the U.S. Having a partner with the same future goals provided Carl with significant mental support:
My girlfriend … is an international student from Cambodia as well …. We’re pretty much there for each other … because we’re really looking forward to our future, especially when we get a job in America. We want to continue our life here, maybe work for the H1-B visa or get a Green Card eventually …. The hope is to stay in America, get a job, [and] build a life here. (Carl, age 23, male, Cambodia)

##### Professor

Having a close relationship with course professors has been beneficial for international students, as they received professional advice on their class performances and career paths. Ben shared how his professors had been supportive throughout his college life. Some of his professors continued to stay in contact with him even after the end of the academic semester, providing guidance on his career goals:
I formed some close bonds with a lot of the faculties and staff from my community college. They are always checking me up…and have been very helpful [in] academic support …. My professor has been the best. She [is] always checking in with me. She’s always giving me professional advice with my career aspirations. (Ben, age 24, male, Myanmar)

In addition to the professional advice international students were able to receive from their course professors, Ben specifically mentioned that one of his professors became his role model. The professor, who helped him with his courses and career planning, had a successful educational background, and Ben expressed a desire to follow in her footsteps. Having a close relationship with and looking up to professors served as a motivational support system for Ben and several other international students:
All the things that you’ve done, she [(professor)] has done …. She has three masters and a PhD from Cornell so she’s my role model. [I] always sought her help whenever I needed it as well, and she has always been supportive …. I want to walk her path. (Ben, age 24, male, Myanmar)

## 4. Discussion

This study is of significance as it represents the first attempt to investigate whether the challenges faced by international students in the U.S. have changed due to the COVID-19 pandemic and how coping strategies have adapted to address these new challenges. Park and Shimada (2022) reported that unstable non-immigrant visa policies during the COVID-19 pandemic negatively impacted international students’ psychological adjustment and well-being [25]. Another study investigated sources of stress or anxiety and coping strategies related to COVID-19 [42]. However, previous research have not fully addressed the unique complexities of chronological challenges and coping strategies from before and during the COVID-19 pandemic. Before the pandemic, international students were already experiencing higher levels of anxiety and depression compared to non-international students [12]. Factors contributing to these discrepancies include limited English proficiency, acculturative stress, feelings of social belonging, and perceived discrimination [43,44]. As the pandemic unfolded, international students faced additional challenges on top of their pre-existing struggles, including emotional stress, the inability to return to home country, and financial difficulties [45]. This study delves into the key findings related to the psychological challenges that international students faced due to an unfamiliar environment before COVID-19, increased mental instability during COVID-19, and the various coping strategies employed both pre-COVID and during the pandemic.

### 4.1. Psychological Challenges Due to Unfamiliar Environment before COVID-19

Before the COVID-19 pandemic, international students faced a myriad of challenges as they embarked on their higher education journey far from home. These challenges encompassed culture shock, language barriers, identity loss, financial struggles, academic challenges, limited university resources, and experiences of racism and discrimination. International students had to adapt to the new environment, often feeling as though they had left a part of their lives behind [46]. With the goal of obtaining higher education, they focused on their studies and work to achieve their objectives. During this adjustment process, international students felt pressured to become self-reliant, having left their families behind in their home countries. Upon starting their higher education in the U.S., they experienced culture shock [47]. Study participants reported feeling nervous about being in a foreign country and having to get used to new laws and regulations. They often felt overwhelmed by the expectation to be accountable for their own lives as they could no longer depend on their family. Racism and discrimination were other common challenges during their adjustment. International students often experience instances of discrimination compared to their native counterparts [47,48]. Likewise, some study participants reported unwelcoming experiences, including racism and discrimination in the form of hate speech. They felt uncomfortable being judged based on their physical appearance and language proficiency. Many international students came from homogenous countries where race is not a significant part of identity. However, upon arriving in the U.S., they were instantly labeled as certain racial categories, leading to discomfort and heightened sensitivity to discrimination [49].

In addition to the challenges faced during the adjustment process, international students had difficulties accessing campus resources, especially those who transferred from a community college to a four-year university. Transferring course credits and receiving inadequate guidance on course planning were the main obstacles for international transfer students [50]. With the goal of pursuing higher education in the U.S., international students spent most of their time on campus. They were required to adapt to various aspects, including the campus environment, social interactions, coursework, instructors, and campus organizations. Failure to make these adjustments often led to stress and anxiety [47]. Regarding academic performance, students were concerned about disappointing their families back home, which added pressure to perform well academically. Low proficiency in English exacerbated these challenges, as non-native speakers struggled with understanding academic terminology. Language barriers also limited communication with professors and other students [32]. Study participants reported a lack of confidence and comfort when speaking English, which hindered their ability to express themselves and voice their opinions. Besides academic challenges, international students encountered financial struggles during their time in the U.S. They had to rely on loans to cover tuition fees, as federal student aid is not available to them due to citizenship restrictions [33]. To manage educational and living expenses, students sought employment; however, their options were limited to on-campus jobs, which were scarce and often provided inadequate pay.

Amidst these challenges, there were positive experiences, such as the offline school setting. Most of the study participants (26/34, 76.47%) were able to experience in-person classes. International students reported enjoying face-to-face interactions and being physically present on campus. They appreciated spending time on campus, meeting new people, and exploring the university. In-person learning before COVID-19 also allowed students to be more engaged in the classroom. Participating in discussions and experiencing unique learning methods, such Socratic seminar, enhanced their enjoyment of studying in the U.S. Offline school helped students make new friends, build friendships, and facilitated communication with peers and professors, allowing them to practice their communication skills.

### 4.2. Increased Mental Instability during COVID-19

During COVID-19, international students continued to face similar challenges as before the COVID-19 pandemic, with some issues becoming more pronounced. Students experienced an increase in mental instability due to an escalated identity crisis, heightened experiences of racism and discrimination, and intensified financial struggles. The escalated identity crisis was linked to a loss of motivation and sense of purpose, along with uncertainty about when campuses would reopen and traveling restrictions would be lifted [51]. International students, especially those of Asian descent, faced increased racial exclusion due to anti-Asian sentiment, which worsened as a result of the pandemic [52,53]. Participants felt unwelcomed, which fostered feelings of inferiority and mistrust towards international students from foreign countries. With campus closures due to pandemic restrictions, international students struggled financially to find job opportunities. As they are generally permitted to work only on-campus jobs, the lockdown severely restricted one of their main sources of income [21]. Furthermore, participants showed concern with the unchanged tuition fees despite their classes being held entirely online. The burden of paying the same tuition and fees while being limited in their social interactions with professors and classmates added to their stress.

In addition to the ongoing challenges during COVID-19, international students experienced new challenges related to online learning and lockdown protocols and showed concerns with their legal status. With all courses shifting to virtual classes, students struggled to maintain effective classroom communications and felt isolated from their communities [54]. This social isolation during the pandemic led to increased reports of worsened mental health symptoms such as anxiety and depression [55]. Furthermore, participants noted that the lack of separation between learning and home environments made it difficult for them to stay focused and motivated. Attending Zoom lectures and completing assignments at home led to academic stress and fatigue, as they had minimal interactions with classmates and felt alone in their efforts to pass the course. Amidst online learning, there was a need for engaging methods to foster enthusiasm and participation in class [56]. Substantial changes to students’ daily routines following the lockdown protocols led to unstable mental health, resulting in anxiety and depression [21,51,54]. The loss of structure in daily life, coupled with the loss of social interactions, contributed to feelings of isolation from the campus community and homesickness due to separation from family and the need for independent coping with the changes. In addition, international students dealt with concerns in maintaining their legal status [25]. The COVID-19 travel restrictions complicated the visa renewal process, and students were anxious about meeting eligibility requirements for re-entry into the U.S.

While international students experienced continuing and new psychological challenges during COVID-19, there were positive aspects of the lockdown, such as the convenience of the online school format. With all courses shifting to a completely online during the pandemic, students living off-campus were able to save time and transportation costs by avoiding early commutes to campus. Furthermore, the availability of class recordings allowed students to review lectures at their own pace, which was particularly beneficial for those with language barriers, especially when professors spoke quickly.

### 4.3. Various Coping Strategies before and during COVID-19

To cope with difficulties while studying in the U.S., international students relied on social connections and community resources and focused on enhancing self-care. Before COVID-19, students managed stress through physical activities such as exercising and exploring new locations [57]. Engaging in physical exercise served as an emotional escape, helping participants clear their minds and relieve the tension caused by life stressors. In addition, students focused on self-improvement, including pursuing hobbies, prioritizing mental self-care, and adjusting their negative mindsets to cope before the pandemic. Maintaining contact with friends and family provided a social support system for international students when coping with stress [47]. While these coping mechanisms continued through the beginning of the pandemic, online communication—video chatting with friends and connecting with their family from the home country—became particularly important for managing stress during the COVID-19 lockdown. The pandemic made it possible for international students to reconnect with their long-distance friends through online communication [58]. Furthermore, participants experienced an elevated motivation for self-improvement by learning foreign languages and cooking. Acquiring new skills helped students stay actively engaged and feel accomplished.

Typical sources of support for international students, regardless of COVID-19, included a supportive community that provided emotional, academic, and social assistance from their family, relatives, and friends [59]. While parental support was often highlighted for providing financial and motivational support, participants expressed their gratitude for their relatives’ contributions during their academic struggles. Many of the participants’ relatives, including their uncles, aunts, and cousins, had been living in the U.S. and had experienced college life outside of their home country. Therefore, students considered their relatives to have been a stronger educational resource compared to their parents. In addition to family support, campus resources such as course professors and international student advising services helped guide students, especially for the first generation students studying in the U.S. [60]. Having access to research assistant opportunities allowed students to feel a purpose in the community, build connections with other research assistants, and adjust quickly to the life in the U.S. After all, international students’ effort to participate in opportunities and contribute to the community needs to be widely recognized [61]. However, despite having access to the campus resources, participants preferred to have had support systems including more approachable campus mental health resources, welcoming campus programs for international students, more paid research opportunities, and be provided with information regarding financial aid, public transportation, and U.S. federal tax information. While managing the stress of living in the U.S., the common mental support systems for the international students with whom they felt closest were their friends, family, partners, and course professors. Friends, both local other international ones, empathized with their struggles and provided advice and resources, such as internship information, during their college life. In addition, participants experienced unconditional emotional support from their family, who had been encouraging of their decision to study abroad and supportive of future aspirations. International students felt their partners were an important mental support system as they shared similar future goals and gave life advice without feeling rejected. Likewise, students considered their course professors as role models who helped drive them toward their future goals.

### 4.4. Limitations

The limitation of this qualitative study, with its small sample size, may affect the generalizability of the findings to the broader population. Variability in psychological impacts and coping strategies among different individuals and contexts could introduce additional limitations. Factors such as geographic location, personal experiences, and cultural backgrounds may differ across states and between individuals, contributing to this variability. Those selected for this study were international students in the San Francisco Area, which focusing on a specific segment of our target population. Variations in COVID-19 policies based on different states could influence the type of challenges international students face. Among the 34 participants, 32 students (94.12%) were Asians with the majority coming from India (8/34, 23.53%) and Vietnam (6/34, 17.65%). While the predominance of Asian international students in this study may reduce generalizability, the overall racial distribution of students in the San Francisco Bay Area aligns with the distribution in our sample. Furthermore, our Asian study participants came from different parts of Asia: West Asia (4/32, 12.5%), East Asia (6/32, 18.75%), South Asia (9/32, 28.13%), and Southeast Asia (13/32, 40.63%). Considering that the San Francisco Bay Area has a substantial population of Asian students, the distribution of participants’ origins is reasonably representative for the study. Therefore, the percentages of participants in our sample can be extrapolated to the general racial composition of international students attending a university in this selected area.

Second, all in-depth interviews we conducted were online, which limited the active human interactions that would have occurred in-person. Participants may have felt uncomfortable sharing their experiences, particularly their emotional challenges, through a screen, compared to having interviewers physically present. To build rapport and gain trust from participants, we made an effort to create a comfortable and safe online environment by offering options such as using a pseudonym and turning off the video. This allowed participants to choose their level of comfort when disclosing personal identifier during the interview.

Finally, although the original intent of this study was to understand the psychological challenges faced by international students during COVID-19, we recognized the value of comparing these psychological challenges and coping strategies before and during the pandemic. This decision stemmed from the inclusion of questions in our original questionnaire about the overall experiences and challenges faced by international students prior to COVID-19. As we collected participant data, our study’s focus shifted to comparing the psychological challenges and coping strategies of international students in the U.S. before and during COVID-19. Our sample included international students enrolled in at least one course at a university in the San Francisco Bay Area from Spring 2020 to Spring 2021. Out of the 34 participants, eight students—Michelle, Patrick, Ernest, Denise, Cheryl, Lawrence, Ralph, and Raymond—arrived during the COVID-19 pandemic, either in Fall 2020 or Spring 2021. Since these eight students did not have the opportunity to experience college life in the U.S. before COVID-19, we were unable to gather their pre-pandemic experiences as international students.

## 5. Conclusions

This study showed that the challenges faced by international students in the U.S. changed with the onset of the COVID-19 pandemic. In addition, there were persistent challenges that were exacerbated by the pandemic. International students contribute to the U.S. economy by bringing diversity to higher education. Therefore, it is critical to analyze and address the underlying struggles these students face, especially with the ongoing challenges due to the COVID-19 pandemic. This study identified common psychological challenges among international students before the pandemic, including academic and financial pressure and difficulty adjusting to the new environment. During COVID-19, students experienced increased mental instability due to the challenges of adapting to pandemic restrictions. Specifically, individuals faced escalated identity crises, leading to a loss of sense of purpose and uncertainty about their futures. Challenges related to online learning, adapting to lockdown protocols, and maintaining their legal status emerged as specific concerns due the pandemic. Despite the COVID-19 pandemic, international students continued to seek support systems, including financial aid, paid research opportunities, and more approachable campus and off-campus resources, highlighting the crucial role these support systems play in student life.

Overall, the challenges faced by international students before the pandemic were worsened by the changes brought about during COVID-19, and new concerns specific to the pandemic were introduced, leading to increased psychological issues. Future studies should suggest effective interventions for international students that support their social and cultural well-being with evidence-based coping skills and strategies for unexpected situations, including disease outbreaks. Additionally, studying the mental health of international students after the COVID-19 pandemic would be valuable for comparing their experiences before, during, and after the pandemic.

## Figures and Tables

**Figure 1 ijerph-21-01232-f001:**
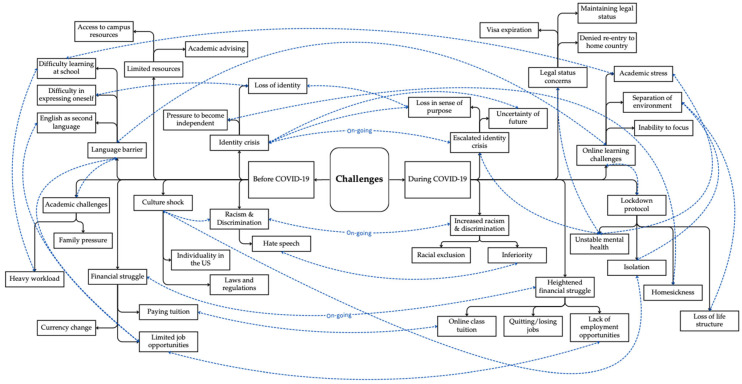
Interrelationship between categories and codes.

**Table 1 ijerph-21-01232-t001:** Participant Demographics (*n* = 34).

Participant Number	Participant Pseudonym	Age	Program	Gender	Country	First Enrolled as College Student in the U.S.	College Student in the U.S.
1	Sean	25	Masters	Male	Lebanon	Before Spring 2020	1–2 years
2	Debra	20	Undergraduate	Female	Ukraine	Before Spring 2020	1–2 years
3	Nicole	25	Undergraduate	Female	South Africa	Before Spring 2020	2–5 years
4	Robert	32	Undergraduate	Male	Thailand	Before Spring 2020	1–2 years
5	Ben	24	Undergraduate	Male	Myanmar	Before Spring 2020	2–5 years
6	Betty	19	Undergraduate	Female	Singapore	Before Spring 2020	1–2 years
7	Shirley	18	Undergraduate	Female	Iraq/UAE	Before Spring 2020	Less than 1
8	Kimberley	24	Undergraduate	Female	Hong Kong/China	Spring 2020	2–5 years
9	Edward	23	Undergraduate	Male	The Philippines	Before Spring 2020	2–5 years
10	Alice	22	Undergraduate	Female	Japan	Before Spring 2020	2–5 years
11	Nancy	25	Masters	Female	Vietnam	Before Spring 2020	2–5 years
12	Carl	23	Undergraduate	Male	Cambodia	Before Spring 2020	5 or more years
13	Lois	25	Masters	Female	Iran	Before Spring 2020	1–2 years
14	Jonathan	22	Undergraduate	Male	Malaysia	Spring 2020	2–5 years
15	Jack	25	Masters	Male	India	Before Spring 2020	1–2 years
16	Michelle	19	Undergraduate	Female	Bangladesh	Spring 2021	Less than 1
17	Eric	20	Undergraduate	Male	Vietnam	Before Spring 2020	1–2 years
18	Evelyn	23	Undergraduate	Female	Japan	Before Spring 2020	2–5 years
19	Earl	26	Masters	Male	India	Spring 2020	1–2 years
20	Stephen	24	Undergraduate	Male	Malaysia	Spring 2020	1–2 years
21	Bruce	21	Undergraduate	Male	Iran	Before Spring 2020	2–5 years
22	Joan	21	Undergraduate	Female	Vietnam	Before Spring 2020	2–5 years
23	William	25	Masters	Male	China	Before Spring 2020	5 or more years
24	Linda	29	Masters	Female	China	Before Spring 2020	2–5 years
25	Patrick	23	Masters	Male	India	Spring 2021	Less than 1
26	Ernest	26	Masters	Male	India	Fall 2020	Less than 1
27	Denise	30	Masters	Female	Taiwan	Fall 2020	Less than 1
28	Cheryl	25	Masters	Female	India	Fall 2020	Less than 1
29	Susan	20	Undergraduate	Female	Vietnam	Before Spring 2020	2–5 years
30	Matt	29	Masters	Male	India	Spring 2020	1–2 years
31	Lawrence	23	Masters	Male	India	Spring 2021	Less than 1
32	Ralph	24	Masters	Male	India	Spring 2021	Less than 1
33	Kelly	23	Undergraduate	Female	Vietnam	Before Spring 2020	1–2 years
34	Raymond	28	Masters	Male	Vietnam	Spring 2021	Less than 1

**Table 2 ijerph-21-01232-t002:** In-depth interview questionnaire.

Domain 1. Before COVID-19	
Q1:	Before COVID-19, how would you describe your overall experience as an international student at your institution? How was your education progressing as an international student in the US?
Q2:	Before COVID-19, What do you think were some unique challenges that international students faced in the United States? What specific challenges did you face?
Domain 2. During COVID-19	
Q3:	How has your living situation changed since COVID-19 began? (location, positively, negatively, etc.)
Q4:	How has your academic performance been impacted by COVID-19? If it has been negatively or positively impacted, what are the reasons for the change?
Q5:	Can you share any other challenges you faced as an international student due to COVID-19?
Domain 3. Coping Strategies	
Q6:	How do you manage the challenges of being an international student? What do you do to reduce stress or for leisure?
Q7:	What type of support system do you have? (Academic, family, friends, etc.)
Q8:	How supportive have your family members been? Are you the first in your family to attend college? How understanding is your family of your experience being a college student?
Q9:	Who is the closest person to you that you share when you are facing challenges? How do they support you to continue your education in the United States?

**Table 3 ijerph-21-01232-t003:** Summary of themes, categories, and codes.

Themes	Sub-Theme	Categories	Codes
Before COVID-19, international students faced the psychological challenges of living against academic and financial pressure and adjusting their lifestyle to an unfamiliar environment.	Challenges	Identity crisis	Loss of identity
			Pressure to grow up and become independent
		Limited resources	Accessibility to campus resources
			Lack of academic advising
		Racism and discrimination	Hate speech
		Language barrier	Difficulty to learn at school
			English as a second language
			Difficulty in expressing themselves
		Academic challenges	Heavy workload
			Family pressure
		Financial struggle	Paying tuition
			Limited job opportunities due to legal status
			Currency change
		Culture shock	Laws and regulations
			Individuality in the US
	Positive experience	Offline school	University community
			Learning opportunities
			Meet new people/make new friends
During COVID-19, international students experienced an increase in mental instability due to the challenges of adapting to the pandemic restrictions and maintaining legal status.	Challenges	Escalated identity crisis	Losing motivation and sense of purpose
			Feelings of uncertainty for the future
		Higher experience of racism and discrimination	Racial exclusion (e.g., Asian hate)
			Inferiority
		Online learning challenges	Lack of separation of learning and home environment
			Inability to stay focused and motivated
			Academic stress/burn out
		Heightened financial struggles	Lack of employment opportunities due to lockdown
			Quitting/losing jobs
			Tuition and fee prices did not change (paying as if still in-person)
		Legal status concerns	Visa expiration
			Fulfilling requirements to maintain legal status
			Fear of denied re-entry (if returning from visiting family in home country)
		Lockdown protocol	Unstable mental health (anxiety and depression)
			Lack of social interaction/Isolation
			Family separation and being independent (homesickness)
			Loss of structure in daily life
	Positive experience	Convenience of online format	No need to commute for class
			Recorded lectures
To cope with the challenges of living in a foreign country before and during COVID-19, international students relied on family and community resources for emotional support and focused on enhancing self-care.	Managing challenges as an international student before COVID-19	Physical activity	Exercise/gym
			Exploring new locations
		Social exposure	Campus involvement
			International friendships
		Self-improvement	Hobbies
			Adjusting mindset
			Time management
		Mental self-care	Meditation
	Changes in managing challenges as an international student due to COVID-19	Increased frequency of online engagement	Group chats/video calls with friends
			Connecting with home country
		Elevated motivation for self-improvement	Learning new foreign languages
			Learning to cook
	Support systems as an international student regardless of COVID-19 (on-going)	Campus resources and job opportunities	Research assistant opportunities
			Department counselors/course professors
			International student services
		Family and friends	Financial support
			Emotional support
			Social support
			Academic support
		Desired support systems	Financial aid for tuition cost
			Approachable campus mental health resources
			Welcoming campus programs/clubs for international students
			Transportation resources
			Paid research opportunities
			US federal information (tax)
	Closest individual for mental support	Friends	Empathize with struggles
			Provide advice and resources (internship information)
		Family	Encouraging of studying abroad
			Supportive of educational aspiration
			Unconditional emotional support
		Partner	Provide advice without rejection
			Share future goals
		Professor	Provide professional advice
			Role model figure

## Data Availability

The participants of this study did not give written consent for their data to be shared publicly, so due to the sensitive nature of the research, supporting data is not available.

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
