# Peer review of "Changes in Psychological Challenges, Positive Experiences, and Coping Strategies among International Students in the United States before and during the COVID-19 Pandemic: A Qualitative Study"

_ijerph, 2024, doi:10.3390/ijerph21091232_

Round 1
Reviewer 1 Report
Comments and Suggestions for Authors
Comments to the Author
The authors of this article did an admirable job on an important topic, aimed to understand how challenges and coping strategies among international students in the United States has changed before and during the COVID-19 pandemic. However, there are several points that require further clarity;
1- Page 2. In your article, you have effectively mentioned the psychological effects of the lockdown. However, to enhance the study's comprehensiveness, it would be valuable to also address the physiological impacts of the deadly COVID-19 infection that necessitated the lockdown. Particularly in the section where you focus on university students, it is crucial to highlight the severe risks that COVID-19 poses to the respiratory system, affecting not only elderly patients but also healthy young individuals, including those who are physically active. This would underscore the seriousness of the infection. It is necessary to bear in mind that psychological effects arise as a result of the containment measures taken by governments to deal with this contagious and deadly disease.
If you are considering such an addition, I suggest consulting the following resources for relevant information.
https://doi.org/10.1002/ejsc.12109
https://doi.org/10.1111/jocn.16352.
2- Page 4. Please indicate that permission for audio recording has been obtained from all participants involved in the study.
GENERAL COMMENTS:
1. The most significant issue in the study is that the findings may not be generalizable. Both the psychological impacts and coping strategies during the pandemic can vary significantly depending on various factors. While the authors have already highlighted that these variations can occur even between states, individual differences may also vary considerably, making it challenging to obtain accurate results from such a small population.
Author Response
Comment 1: The authors of this article did an admirable job on an important topic, aimed to understand how challenges and coping strategies among international students in the United States has changed before and during the COVID-19 pandemic. However, there are several points that require further clarity;
Respond 1: We would like to appreciate your time to review our manuscript. Thank you very much for providing positive comments.
Comment 2: 1- Page 2. In your article, you have effectively mentioned the psychological effects of the lockdown. However, to enhance the study's comprehensiveness, it would be valuable to also address the physiological impacts of the deadly COVID-19 infection that necessitated the lockdown. Particularly in the section where you focus on university students, it is crucial to highlight the severe risks that COVID-19 poses to the respiratory system, affecting not only elderly patients but also healthy young individuals, including those who are physically active. This would underscore the seriousness of the infection. It is necessary to bear in mind that psychological effects arise as a result of the containment measures taken by governments to deal with this contagious and deadly disease.
If you are considering such an addition, I suggest consulting the following resources for relevant information.
https://doi.org/10.1002/ejsc.12109
https://doi.org/10.1111/jocn.16352.
Respond 2: Thank you for your suggestion to include the physiological impacts of COVID-19 infection and for providing us with two relevant resources. We have included the two articles that you suggested and added the following sentences:
In response to the pandemic restrictions, students faced an increase in both physiological and psychological risks. COVID-19 presented severe risks to not only elderly patients but also to healthy young individuals, including students who were physically active [17]. Symptoms such as fatigue, muscle weakness, and reduced respiratory function associated with COVID-19 infection were linked to an altered psychological state among college students, including post-traumatic stress disorder (PTSD), anxiety, and depression [17,18].
References:
17. Karaduman, E.; Bostancı, Ö.; Bilgiç, S. A Follow-up Study on Respiratory Outcomes, Quality of Life and Performance Perception of SARS-CoV-2 Primary and Reinfection in Elite Athletes: A 9-Month Prospective Study. EJSS (Champaign) 2024, 24, 964–974.
18. Del Corral, T.; Menor-Rodríguez, N.; Fernández-Vega, S.; Díaz-Ramos, C.; Aguilar-Zafra, S.; López-de-Uralde-Villanueva, I. Longitudinal Study of Changes Observed in Quality of Life, Psychological State Cognition and Pulmonary and Functional Capacity after COVID-19 Infection: A Six- to Seven-Month Prospective Cohort. J. Clin. Nurs. 2024, 33, 89–102.
Comment 3: 2- Page 4. Please indicate that permission for audio recording has been obtained from all participants involved in the study.
Respond 3: Thank you! We have added the following sentence in 2.6. Ethical considerations:
Permission for audio recording has been obtained from all participants involved in the study.
Comment 4: GENERAL COMMENTS:
1. The most significant issue in the study is that the findings may not be generalizable. Both the psychological impacts and coping strategies during the pandemic can vary significantly depending on various factors. While the authors have already highlighted that these variations can occur even between states, individual differences may also vary considerably, making it challenging to obtain accurate results from such a small population.
Respond 4: We agree that a qualitative study in general may not be generalizable to the entire population. We strengthened this generalizability concept at the beginning of 4.4. Limitations as follows:
The limitation of this qualitative study, with its small sample size, may affect the generalizability of the findings to the broader population. Variability in psychological impacts and coping strategies among different individuals and contexts could introduce additional limitations. Factors such as geographic location, personal experiences, and cultural backgrounds may differ across states and between individuals, contributing to this variability.
Reviewer 2 Report
Comments and Suggestions for Authors
It is a good piece of qualitative research on an interesting topic. I suggest adding some more references to the discussion.

Comments on the Quality of English LanguageThe manuscript needs extensive proofreading.
Author Response
Comment 1: It is a good piece of qualitative research on an interesting topic. I suggest adding some more references to the discussion.
Respond 1: We would like to appreciate your time to review our manuscript. Thank you very much for providing positive comments.
We have added the references in the discussion with additional explanations below:
Discussion 4.1
“International students often experience instances of discrimination compared to their native counterparts.”
- Zhang, Q., Xiong, Y., Rose Prasath, P., & Byun, S. (2023). The relationship between international students’ perceived discrimination and self-reported overall health during COVID-19: Indirect associations through positive emotions and perceived social support. Journal of International Students, 14(1), 119–133. https://doi.org/10.32674/jis.v14i1.5368
“Many international students came from homogenous countries where race is not a significant part of identity. However, upon arriving in the U.S., they were instantly labeled a certain racial categories, leading to discomfort and heightened sensitivity to discrimination.”
- Loo, B. (2019). International Students and Experiences with Race in the United States. World Education News + Reviews. https://wenr.wes.org/2019/03/international-students-and-experiences-with-race-in-the-united-states
“With the goal of pursuing higher education in the U.S., international students spent most of their time on campus. They were required to adapt to various aspects, including the campus environment, social interactions, coursework, instructors, and campus organizations. Failure to make these adjustments often led to stress and anxiety.”
- Kristiana, I. F., Karyanta, N. A., Simanjuntak, E., Prihatsanti, U., Ingarianti, T. M., & Shohib, M. (2022). Social Support and Acculturative Stress of International Students. International journal of environmental research and public health, 19(11), 6568. https://doi.org/10.3390/ijerph19116568
Discussion 4.2
“International students, especially those of Asian descent, faced increased racial exclusion due to anti-Asian sentiment, which worsened as a result of the pandemic.”
- Zhang, X., Hsu, K. C., Fleming, K. E., Liu, C. H., & Hahm, H. C. (2023). Home away from home: international students' experiences during the COVID-19 pandemic and the role of US higher education. Frontiers in psychology, 14, 1104200. https://doi.org/10.3389/fpsyg.2023.1104200
“This social isolation during the pandemic led to increased reports of worsened mental health symptoms such as anxiety and depression.”
- Trusty, W. T., & Chun-Kennedy, C. (2024). International Students Are More Socially Isolated than Domestic Students, and the Gap Is Growing After COVID-19. PennState Student Affairs Center for Collegiate Mental Health. https://ccmh.psu.edu/index.php?option=com_dailyplanetblog&view=entry&year=2023&month=09&day=07&id=44:international-students-are-more-socially-isolated-than-domestic-students-and-the-gap-is-growing-after-covid-19#
“Amidst online learning, there was a need for engaging methods to foster enthusiasm and participation in class.”
- Gherghel, C., Yasuda, S., & Kita, Y. (2023). Interaction during online classes fosters engagement with learning and self-directed study both in the first and second years of the COVID-19 pandemic. Computers & education, 200, 104795. https://doi.org/10.1016/j.compedu.2023.104795
Discussion 4.3
“The pandemic made it possible for international students to reconnect with their long-distance friends through online communication.”
- Thorson, A. R., Doohan, E. A. M., & Clatterbuck, L. Z. (2022). Living Abroad During COVID-19: International Students’ Personal Relationships, Uncertainty, and Management of Health and Legal Concerns During a Global Pandemic. Journal of International Students, 12(3), 654—673. https://doi.org/10.32674/jis.v12i3.3613
“After all, international students’ effort to participate in opportunities and contribute to the community needs to be widely recognized.”
- Helms, R. M., & Spreitzer, S. (n.d.). International Student Inclusion and Success: Public Attitudes, Policy Imperatives, and Practical Strategies. American Council on Education. https://www.acenet.edu/Documents/International-Student-Inclusion-Success.pdf
Comment 2: The manuscript needs extensive proofreading.
Respond 2: Thank you very much for taking the time to proofread our manuscript. We have incorporated all your comments into the revised version. Additionally, we wanted to highlight that we aimed for verbatim transcription to present participants’ quotes as accurately as possible, even if they contained grammatical errors. However, we agreed that removing filler words and correcting minimal grammar errors with square brackets were necessary in the participants’ quotes, and we have made those adjustments accordingly.
Reviewer 3 Report
Comments and Suggestions for Authors
Dear authors. Congratulations on your research and on the subject matter of it. Without detracting from the subject matter, you should bear in mind that your article would have been more interesting in previous years. As for the design of the study, there are no clear objectives in relation to the questions posed to the research subjects. This is also missing from the conclusions, which I recommend you delve into a little more. As for the data analysis, in Table 2, it would be necessary to indicate the relationship of the codes with the highest index of rooting and density, as well as the relationship between codes. This last graph would give us more detailed information and would facilitate the understanding of your research. In addition, it would give us more reliability of the data. Finally, I recommend that you review the bibliographical references, they do not match in format. For example, citations 26, 47 and 48, among others, do not have a year of publication.

Author Response
Comment 1: Dear authors. Congratulations on your research and on the subject matter of it. Without detracting from the subject matter, you should bear in mind that your article would have been more interesting in previous years.
Respond 1: We would like to appreciate your time to review our manuscript. Thank you very much for providing positive comments.
Comment 2: As for the design of the study, there are no clear objectives in relation to the questions posed to the research subjects.
Respond 2: Our in-depth interview questionnaire was designed with a conversational tone, but we have made slight grammatical adjustments to clarify each question.
In addition, we have added clear objectives in the 2.1. Study Design section as follows:
The in-depth interviews were structured around three main objectives. First, we aimed to understand the pre-pandemic experiences of international students, including their overall experience at their institution and the specific challenges they faced. This helped establish a baseline for comparison with during the pandemic. Second, we examined the impact of COVID-19 on students’ living situations, academic performance, and additional challenges they encountered. This aimed to highlight how the pandemic has altered their experiences. Third, we identified the coping strategies and support systems that they use to manage these challenges. This included understanding how they handle stress and which sources of support are most effective for them.
Comment 3: This is also missing from the conclusions, which I recommend you delve into a little more. As for the data analysis, in Table 2, it would be necessary to indicate the relationship of the codes with the highest index of rooting and density, as well as the relationship between codes. This last graph would give us more detailed information and would facilitate the understanding of your research. In addition, it would give us more reliability of the data. Finally, I recommend that you review the bibliographical references, they do not match in format. For example, citations 26, 47 and 48, among others, do not have a year of publication.
Respond 3: We have addressed them in the next rows.
Comment 4: Pg 4. To improve the interest of the study, I recommend collecting data after COVID-19.
Respond 4: Thank you very much for this invaluable suggestion. However, our qualitative study was designed as a one-time, one-on-one in-depth interview, which did not include follow-up with participants. We aimed to highlight the challenges and coping strategies of international students specifically before and during COVID-19. Because collecting data after COVID-19 requires a new study with new IRB approval, we would like to conduct another study to understand mental health among international students after the COVID-19 pandemic in the near future.
We have added the following sentence in the Conclusions section to highlight the future study:
Future studies should suggest effective interventions for international students that support their social and cultural well-being with evidence-based coping skills and strategies for unexpected situations, including disease outbreaks. Additionally, studying the mental health of international students after the COVID-19 pandemic would be valuable for comparing their experiences before, during, and after the pandemic.
Comment 5: Pg 5. I recommend including more information about the codes, for example
the relationship between codes with the highest rooting index and density.
Respond 5: Thank you for your recommendation. The categories and codes from Table 2 are organized in order of highest rooting index (the higher the frequency of participants describing the codes, the higher in the list in each category). We have specified this as follows:
The categories and codes from Table 2 are organized from highest to lowest rooting index and density. Categories more frequently described by participants within the sub-themes are listed first in the table. Likewise, codes are listed in the order of density within their respective categories.
Comment 6: Pg 7. Results. I recommend including relationship graphs between codes.
Respond 6: We appreciate this suggestion. We have added Figure 1, which illustrates the interrelationship between categories and codes. In addition, we have added the following sentences to describe the figure:
Figure 1 illustrates the interrelationship between categories and codes of challenges both before and during the COVID-19 pandemic. Solid lines indicate the codes and categories for each sub-theme, while dotted lines connect codes that were frequently mentioned together in the in-depth interviews. Specifically, three categories were consistently identified as ongoing challenges: identity crisis, racism and discrimination, and financial struggle.
Comment 7: Pg 30. Conclusions. I recommend drawing more concrete conclusions in relation to the stated purposes.
Respond 7: Thank you for this recommendation. We have added the following sentences in the Conclusions:
Specifically, individuals faced escalated identity crises, leading to a loss of sense of purpose and uncertainty about their futures. Challenges related to online learning, adapting to lockdown protocols, and maintaining their legal status emerged as specific concerns due the pandemic. Despite the COVID-19 pandemic, international students continued to seek support systems, including financial aid, paid research opportunities, and more approachable campus and off-campus resources, highlighting the crucial role these support systems play in student life.
Overall, challenges faced by international students before the pandemic were worsened by the changes brought about during COVID-19, and new concerns specific to the pandemic were introduced, leading to increased psychological issues.
Comment 8: Pg 31. References. They need to be reviewed. Data is missing. For example:
26. Marbang, McKinzie, Eller, Leggett. International Students' Experiences with
Changing Policy: A Qualitative Study from Middle Tennessee. J interdiscip multidiscip res.
Respond 8: We have updated this reference with the publication year:
Marbang, P.; McKinzie, A.E.; Eller, J.L.; Leggett, I.F. International Students’ Experiences with Changing Policy: A Qualitative Study from Middle Tennessee. JISE 2020, 9, 301–329.
Comment 9: 48. Koo KK, Yao CW, Gong HJ. "It is not my fault": Exploring experiences and perceptions of racism among international students of color during COVID-19. J Divers High Educ. 16:284-96.
Respond 9: We have updated this reference with the publication year:
Koo, K.K.; Yao, C.W.; Gong, H.J. “It Is Not My Fault”: Exploring Experiences and Perceptions of Racism among International Students of Color during COVID-19. J. Divers. High. Educ. 2023, 16, 284–296.
Comment 10: 49. Martirosyan NM, Van De Walker D, Patrick Saxon D. The Impact of the COVID-19 Pandemic on International Students in a Public University in the United States: Academic and Non-academic Challenges. JCIHE. 2022;14:90--102.
Respond 10: We have checked this reference, and it includes the publication year:
Martirosyan, N.M.; Van De Walker, D.; Patrick Saxon, D. The Impact of the COVID-19 Pandemic on International Students in a Public University in the United States: Academic and Non-Academic Challenges. JCIHE 2022, 14, 90–102.